# Peer Status as a Potential Risk or Protective Factor: A Latent Profile Analysis on Peer Status and Its Association with Internalizing Symptoms in Adolescents with and without Parental Physical Abuse Experience

**DOI:** 10.3390/children9050599

**Published:** 2022-04-22

**Authors:** Céline A. Favre, Dilan Aksoy, Clarissa Janousch, Ariana Garrote

**Affiliations:** Department of Research & Development, School of Education, University of Applied Sciences and Arts Northwestern Switzerland, 5210 Windisch, Switzerland; dilan.aksoy@fhnw.ch (D.A.); clarissa.janousch@fhnw.ch (C.J.); ariana.garrote@fhnw.ch (A.G.)

**Keywords:** peer status, parental physical abuse, internalizing symptoms, peer acceptance, peer rejection, popularity, latent profile analysis

## Abstract

Research has well established that parental physical abuse experiences can lead to devastating consequences for adolescents, with peer relationships acting as both protective and risk factors. With the person-centered latent profile analysis (LPA), we analyzed questionnaire data from a cross-sectional study in 2020 composed of a sample of 1959 seventh-grade high school students from Switzerland. This study investigated and compared peer-status profiles combining peer acceptance and peer popularity for adolescents with and without parental physical abuse experiences. We conducted a multinomial logistic regression analysis to investigate further depression, anxiety, and dissociation as predictors of profile membership. With LPA, we identified three distinct profiles for adolescents within the subgroup with experiences of parental physical abuse (*n* = 344), namely liked, liked-popular, and rejected-unpopular. Within the subgroup of adolescents without parental physical abuse experiences (*n* = 1565), LPA revealed four profiles, namely liked, liked-popular, rejected-unpopular, and average. For adolescents with parental physical abuse experiences, higher levels of dissociation significantly indicated they were more likely to belong to the rejected-unpopular group than belong to the liked group. Anxious students without experiences of parental physical abuse were more likely to belong to the rejected-unpopular and liked profiles than belong to the liked-popular and average profiles. These findings clearly argue for a deeper understanding of the role of parental physical abuse when analyzing the relationship between dissociation and anxiety and peer status. Operationalizing peer status with the four individual dimensions of likeability, rejection, popularity, and unpopularity was valuable in that the role of peer rejection with respect to different internalizing symptoms became apparent.

## 1. Introduction

Research has shown that parental abuse is a common burden for youth [1,2]. In Switzerland, approximately 19% of youth are exposed to parental physical abuse [3], in the European Union around 20–25% [2,3,4] and 18% of American youth experience parental physical abuse at least once in their lifetime [5]. Parental abuse, also called child maltreatment, can take on different forms, including parental physical abuse being and inflicting nonaccidental bodily injury. In meta-analyses, Evans et al. [6], Kitzmann et al. [7], and Lindert et al. [8] show significant evidence that exposure to parental abuse leads to a range of negative psychosocial outcomes in adolescence, in particular an increase in internalizing symptoms, such as depression [9,10,11], anxiety [12,13], and dissociation [14,15].

How adolescents respond to such adverse abuse experiences can be understood in multisystemic terms [16]. That is, whether a person embedded in interdependent systems has the capacity to adapt successfully to adversity and therefore shows resilience [17]. Resilience in the context of abuse concerns individuals who, despite histories of abuse and thus increased risk for developing internalizing and externalizing symptoms, do not exhibit negative developmental trajectories [18]. In the context of adolescents’ experiences of abuse and resilience, peers play an important role. Peer acceptance acts as a key protective factor that can prevent psychopathological symptoms [15,19,20]. However, the peer group can also increase the risk of rejection due to dysregulated behaviors of adolescents with and without parental physical abuse experiences as another risk factor in the social environment [21]. Peer rejection, as a dimension of peer acceptance, increases the attribution of hostile intentions for others’ behavior, decreases the development of competent solutions to interpersonal situations [22], and it can be an additional risk factor for healthy development. In this context, peer acceptance importantly indicates resilient adaptation to adversity [23,24].

With the frequency and intensity of peer relationships increasing as children enter adolescence [25], peer relationships begin to play a crucial role in cognitive and emotional development [26,27]. Studies examining the relationship between parental abuse and peer status primarily show that abuse leads to higher levels of peer rejection [28,29,30] and lower levels of peer acceptance [31,32,33,34]. This can be inferred from the fact that abuse influences how someone behaves in the peer group. Bolger and Patterson [23] found a causal link between abuse, dysregulated behavior toward others, and resulting peer rejection at an early school age that persists into early adolescence.

Most research focuses on studies of externalizing behaviors in relation to peer status, and only a few studies address internalizing symptoms related to peer status [35].

A basic approach to identifying and studying the resources and protective factors associated with resilience is a person-centered analysis. Studies have compared groups of people who meet certain criteria for risk and positive adjustment with other groups who either have the same risk but are poorly adjusted or have the same positive outcomes but are at lower risk [17].

The present study, with its large sample of participants, combines both approaches by using a person-centered latent profile analysis (LPA) to examine how a group of adolescents with parental physical abuse experiences and their peer status are associated. We aim to discover whether patterns regarding peer status can be identified in adolescents with and without abuse experiences; that is, whether they can be assigned to homogeneous peer-status profiles. Another goal is to examine internalizing symptomatology, such as depression, anxiety, and dissociation, in the adolescents with and without physical abuse experiences and examine whether these symptoms relate to the peer status patterns that have been identified. A downward spiral can occur in the reciprocal relationship among adolescent behavior, internalizing symptoms, and peer rejection [36]. Furthermore, the same procedure in both adolescent groups enables a comparison between the profiles with and without abuse experiences. Moreover, most studies addressing peer status use the approach of classifying adolescents into status groups with cut-off values or combine various dimensions, such as combining likeability and rejection to peer preference. The present study does not use cut-off values or combinations of the four dimensions of likeability, rejection, popularity, and unpopularity. This is because subtle nuances could be lost in combining indicators; in particular, peer rejection seems to be an important indicator and should stand alone. As a first step, following van den Berg, Burk, and Cillessen [37], this study uses a person-centered approach to understand peer-status profiles in their complexity using four dimensions with and without parental physical abuse experiences. As a second step, this paper investigates whether internalizing symptoms relates to membership in the respective peer-status profiles that LPA can identify.

### 1.1. Peer Acceptance and Popularity as Two Distinct Aspects of Youths’ Peer Status

Peer status reflects each individual’s social position within their social group and is a multidimensional construct [38]. As Mayeux et al. [39] pointed out, popularity was originally described as the peer group generally accepting an individual and was associated with positive attributes attached to status (e.g., with prosocial behavior and low levels of aggression). Coie et al. [40] were the first to present five sociometric status categories for adolescents that sociometric methods assessed: popular, average, rejected, neglected, and controversial. In the late 1990s, Parkhurst and Hopmeyer [41], as well as La Fontana and Cillessen [42], distinguished between sociometric popularity—most liked by peers—describing “popular” through Coie et al.’s [40] five sociometric status categories and reputation-based popularity. Sociometric popularity referred to positive attributes and nonaggressive behavior, while reputation-based popularity linked to both positive and negative attributes. Since then, research has started to focus on two forms of higher status: peer acceptance based on likeability and rejection and reputation-based popularity [39], which henceforth will be called popularity. Although related, popularity and peer acceptance are two unique and distinct peer status dimensions [39,43,44]. Popularity reflects visibility and being an influential peer group member [43,45], while acceptance refers to peers liking an individual more than disliking them [40]. The operationalization of these two status forms is applied differently in peer relationship research, and thus it leads to varying results. To measure peer acceptance, Cillessen and Marks [46] suggested including explicitly both likeability and rejection as two separate indicators of peer acceptance. Marks et al.’s [47] recent findings on popularity similarly showed popularity has in fact two dimensions and should be measured separately through popularity and unpopularity. To capture these four constructs, so-called computer-based unlimited peer nominations have proven to be best for large samples [46]. Based on Coie et al. [40], who found that likeability and peer rejection are not opposite ends of the same continuum, and to follow Marks et al.’s [47] recommendations for likeability and rejection, we conclude it is methodologically useful for our research questions to measure separately the four sociometric dimensions.

### 1.2. Person-Centered Approach in Peer-Status Research

Most peer-status studies use the Coie, Dodge, and Coppotelli [40] (CDC) approach to create status categories, which is based on computed subjective cut-off values [48]. Although increasingly used in peer-relationship research, person-centered approaches, such as LPA, are still understudied in peer-status research. However, a few studies have used them to construct peer status [44]. For example, Hubbard et al. [49] showed in their study that although there was a group of rejected children with the CDC approach as well as with an LPA, these groups differed regarding rejection from each other, and considerably more children were in the rejected category according to CDC than in the rejected LPA group. Van den Berg et al. [44] highlighted in their meta-analysis that the distinction between popular and likeable groups of high-status adolescents in the early years of secondary school was only found in studies using person-centered approaches, whereas in other analytical approaches, this distinction was only found with increasing age. This shows that person-centered approaches are useful in finding specific groups of youth who would otherwise not be found.

### 1.3. Influence of Parental Physical Abuse on Adolescents’ Peer Status in the Context of Resilience

Children exposed to parental abuse have problems developing healthy peer relationships, leading to low popularity and peer group rejection [35,50]. In 2020, as one of the few researchers who adapted a mixture model for peer status, Yoon examined peer dynamics and peer popularity using a latent class analysis to explore whether the profiles of peer relationships differed based on type of abuse. Her results showed that adolescents who experienced parental physical abused were more likely to be ignored by their peers, compared to other types of parental abuse, whereas popularity did not clearly discern the differences between the latent classes in her study. Furthermore, Wang [34] showed that harsh parenting (including physical abuse) negatively related to peer acceptance.

As research has shown, parental abuse in childhood and adolescence increases the risk of externalizing (e.g., peer aggression) and internalizing (e.g., depression) behaviors. Peer status, in turn, further influences psychopathological outcomes because peer acceptance can act as a protective factor, and peer rejection may serve as a risk factor for healthy development [51]. In addition, studies have shown that peer rejection–as a fairly stable process–reduces peer trust in girls and perceived peer support in boys [52].

In a study with young children, Anthonysamy and Zimmer-Gembeck [53] found that children with a history of abuse (physical abuse included) were significantly rejected more compared to their non-abused classmates, and their teacher described them as more physically/verbally aggressive, more withdrawn, and less prosocial than their non-abused peers. The study showed that maltreated children’s behavior mediated the association between maltreatment and peer status. This indicated that maltreated children showed more negative and less positive behaviors toward their peers, leading to more rejection and less likeability nominations.

Individuals who develop adaptively despite challenging or threatening circumstances are said to be on a resilient pathway [54]. Acceptance from peers is an important developmental task for adolescents and an indicator of healthy development [24]. Peer acceptance not only affects self-esteem [55] but also protects it from the negative effects of limited closeness to parents, suggesting that peer acceptance can be a particularly valuable source of self-esteem when closeness to parents is low [56]. Another putative indicator of adaptive development is popularity, which has been associated with low risk for psychological maladaptive development and high social competence. However, recent studies have shown that positive behaviors did not solely describe popularity, but popularity was also positively associated with aggressive and disruptive behavior and negatively associated with prosocial and academic behavior. On the other hand, acceptance is positively associated with prosocial and academic behavior and not significantly associated with aggressive or disruptive behavior [57].

### 1.4. Relationships between Internalizing Symptoms and Peer Status

Coyne [58] developed the interactional model based on interpersonal theory, one of the most influential models focusing on peers’ interpersonal responses to internalizing symptoms. He assumed that interpersonal behavior of people with internalizing symptoms produces rejection from others. Only a few studies thus far have highlighted the link between internalizing symptoms and peer status. For example, Hubers et al. [59] demonstrated a significant association among popularity, acceptance, and internalizing symptoms in older adolescents. In their review, Prinstein et al. [35] highlighted a reciprocal association between negative social experiences within the peer group and internalizing symptoms. However, Mori [60] showed that the path from peer relationship problems to dissociation had a smaller effect size compared to the path from dissociation to peer relationship problems. Thus, there is an indication that internalizing symptomatology may well affect peer relationships. The following sections highlight the established links between peer status and internalizing symptoms, such as depression, anxiety, and dissociation.

#### 1.4.1. Depression and Peer Status

Few studies in the literature have explored the predictive effect of depression on peer status. In a video-based study, Peterson et al. [61] generated evidence that peer rejection occurred in reaction to depressive symptoms in children grades 3 to 6. Peers rated depressed children as less likeable than nondepressed children. Kennedy et al. [62] found evidence indicating that depression was associated with decreases in peer status, as they reported lower peer acceptance levels for depressed primary school-aged children. In a recent study, Malamut et al. [63] examined the association between depressive symptoms and subsequent negative peer experiences (unpopularity and rejection) among adolescents in a gang context. Peer rejection did not predict depression, but depressive symptoms significantly predicted boys’ unpopularity but not that of girls. Thus, it appears that on the one hand, depression can lead to interpersonal problems, such as peer rejection, but also that interpersonal problems often result in depression. This finding was not only evident but core in Platt et al.’s [64] study, which identified peer rejection as a particularly important source of stress. They demonstrated that existing studies showed a bidirectional relationship between peer rejection and depressive symptoms that could influence the development and maintenance of depression.

#### 1.4.2. Anxiety and Peer Status

Many studies have demonstrated that adolescents who suffered from abuse were at higher risk of exhibiting anxiety symptoms [65,66,67]. As a further indicator of internalizing symptoms, high levels of anxiety in adolescence have also been linked to poor peer status, such as high levels of peer rejection [68]. Among anxiety disorders, social anxiety is the most common form of internalizing symptoms in adolescence [69,70]. Therefore, it is not surprising that the interaction between anxiety symptoms and avoidance of close peer relationships likely plays a role in aggravating anxiety and difficulties in peer status [71]. For example, Inderbitzen et al. [72] examined whether adolescents with social anxiety were liked or rejected. The results showed that rejected adolescents displayed increased social anxiety compared to those who were rated as liked, average, or controversial. These results are also consistent with findings from de Lijster’s [73] systematic review, which indicated that higher levels of social anxiety led to less peer acceptance. Further, De Matos et al.’s [74] study of adolescents found that adolescents who had symptoms of both depression and anxiety showed a lower peer status.

However, some studies report different results. For example, Baartmans et al. [75] showed that children with higher social anxiety perceived that their classmates liked them less, but that their peers were less likely to reject them than children with lower levels of social anxiety.

#### 1.4.3. Dissociation and Peer Status

Dissociation is the absence of the integration of thoughts, feelings, and experiences into the stream of consciousness [76]. In extreme situations, such as during physical abuse experiences, dissociation becomes a survival tool to navigate overwhelming feelings [77]. Farina and Liotti [78] reported that early trauma contributes to the development of dissociation, which in turn can lead to psychopathological vulnerability. In particular, parental physical abuse proved to significantly predict the development of dissociation at the clinical level [18]. In adolescence, dissociation can be associated with emotive–relational and behavioral difficulties, such as peer relationship problems [60,79]. Victimized youths more likely have difficulty forming positive and stable relationships with peers. This can be attributed in part to trauma-related problems that may affect the child’s ability to engage successfully in age-appropriate tasks or activities, and trance-like states may be noticeable to other peers and may be judged as strange or uncooperative [80]. In a recent study, Mori [60] found evidence that dissociation predicted peer relationship problems. Thus, dissociation in adolescence likely increases the vulnerability to relationship difficulties. Peer rejection has been linked to dissociative symptoms in children after adverse experiences [81]. However, it is still mostly unknown whether dissociation is related to peer status.

### 1.5. Sociodemographic Variables and Peer Status

In their meta-analysis, van den Berg, Lansu, and Cillessen [44] showed that the association between peer acceptance and popularity only differed among older adolescents. The correlation was weaker for girls than it was for boys. This may be related to the fact that popular girls tend to be less liked because they incur more costs of likeability for popular status than boys do despite the same behavior. It was assumed that older adolescents already developed an awareness of gender norms for niceness (female norm) and dominance (male norm). Increased awareness of these norms related to how adolescents evaluated their female peers in central positions, and they saw influential and popular females as less likeable than males in the same positions. This indicates that likeability and popularity are different constructs because adolescents who are well liked may not necessarily also be popular, influential, and powerful [41,42]. In relation to the experience of abuse, studies show that gender in early adolescence does not seem to play a role in the relationship between peer acceptance and popularity [23] or peer acceptance and abuse [82].

Furthermore, research findings show mixed results for the influence of migration background and socioeconomic status on peer status. Alivernini et al. [83] demonstrated that peers accepted youths with immigrant backgrounds and low socioeconomic status less. On the other hand, Kovacev and Shute [84] identified that adolescents with a migration background received high peer acceptance values, especially if they had a positive attitude toward heritage and host cultures. Regarding popularity, Stevens et al.’s [85] study showed that youth with migration backgrounds were more popular compared to their native classmates.

Regarding socioeconomic status, a positive relationship was found to peer status. Bukowski et al. [86] found in their review that all peer-assessed characteristics (e.g., peer acceptance and popularity) were more pronounced among upper-middle-class youth compared to lower-middle-class youth.

### 1.6. Current Study

Looking at the research to date, we identified important aspects concerning the relationship between parental physical abuse of youth and peer status that have thus far been neglected and were incorporated in the underlying study. The present study conceptualized peer status as profiles based on acceptance and popularity measures, which builds on Coie et al.’s [40] original concept. Following van den Berg and colleagues [37], a person-centered approach was used to understand peer-status profiles in their complexity using likeability and rejection (dimensions of peer acceptance) and popularity as well as unpopularity as separate indicators. Peer rejection alone, and as an indicator of peer acceptance, plays an important role in adolescents’ healthy development. Therefore, it is important to consider the individual dimensions (likeability, rejection, popularity, and unpopularity) of peer status without cut-off values to determine, from the perspective of resilience theory, which adolescents who have experienced abuse are on a resilient pathway regarding peer relationships and benefit from positive peer status, and which become more vulnerable because of peer relationships.

Studies show a strong link between parental abuse and internalizing symptoms [18,87,88] as well as an association between internalizing symptoms and poor peer status [35]. This indicates that parental abuse relates to dysregulated behavior in the peer context and therefore relates to position in the peer group. Still, only few studies have examined peer status in conjunction with parental physical abuse and internalizing symptoms, e.g., [87,89]. Internalizing symptoms are mostly considered as outcomes of poor peer relationships, although there is a strong association of youth with abuse experiences and higher internalizations, e.g., [90]. Therefore, internalizations should not be considered solely as an outcome but also as a predictor. Following the interpersonal theories of internalizing symptoms as a reciprocal association between negative social experiences within the peer group and internalizing psychopathology, e.g., [35], various internalizing symptoms were treated as predictors of peer status profile membership.

Thus, to compare adolescents with parental physical abuse experience and adolescents without physical abuse experience in order to elicit peer-status profiles, we investigated the following three exploratory research questions (RQs) and hypotheses (Hs):

**Research** **Question** **1** **(RQ1).**What peer-status profiles can be found for adolescents with and without parental physical abuse experiences?

**Hypothesis** **1** **(H1).**
*Based on van den Berg et al.’s [37] findings, we hypothesized that at least three profiles would be found: rejected-unpopular, liked-popular, and average.*


**Research** **Question** **2** **(RQ2).**Are there differences in the underlying profiles of peer status between adolescents with and without parental physical abuse experiences?

**Hypothesis** **2** **(H2).**
*We expected differences between the profiles for the subgroups with and without parental physical abuse experiences, based on the findings that a higher proportion of adolescents who experience parental physical abuse are rejected and less often liked by their peers compared to adolescents who do not experience parental physical abuse, e.g., [53,89].*


**Research** **Question** **3** **(RQ3).**How do different forms of internalizing symptoms (i.e., depression, anxiety, and dissociation) predict the memberships of these underlying peer-status profiles?

**Hypothesis** **3** **(H3).**
*According to several research findings, e.g., [60,73,91], we hypothesized that different forms of internalizing symptoms (depression, anxiety, and dissociations) would predict membership in adolescent peer-status profiles.*


## 2. Materials and Methods

### 2.1. Sample

The data analyzed in this research derive from a cross-sectional sample of a broader study on adolescents’ resilience from violence despite experiencing family violence. This study was conducted in autumn 2020. The random sample consisted of 1974 seventh-grade high school students (12–13 years old) from Switzerland, consisting of 1000 (51.2%) assigned females and 952 (48.8%) assigned males, who anonymously completed the online questionnaire in their classroom. We obtained signed consent forms from the students and their parents without an incentive. The ethics committee of the University of Zurich, Switzerland, authorized this project. On the day of the study, the research team members gave a brief oral introduction of the study to participating adolescents of the 140 participating classes, after which the participants completed the questionnaire in about 60 min. The mean age of the total sample was M = 11.76 (SD = 0.65). Of the participating adolescents, 1029 (52.6%) were Swiss citizens and 945 (48%) had a migration background. The main nationalities in Switzerland are 52.6% Swiss, 37.4% other European, and 10% other.

### 2.2. Measures

#### 2.2.1. Grouping Variable

Parental physical abuse was assessed using five items from the Alabama Parenting Questionnaire [92]. The two dimensions, physical aggression and corporal punishment, were assessed, with a focus on severe parental physical abuse. A five-point Likert scale ranging from 1 = never to 5 = always was used (Cronbach’s α = 0.83). The scale included items such as, “My parents beat me so badly that I had to see a doctor or rush to the hospital” and “My parents hit me with a belt, a stick, or a hard object when I did something wrong.” For the LPA, the scores were dichotomized, 1 = never = 0 and >1 = yes-parental physical abuse experience = 1.

#### 2.2.2. Indicators

Peer status. Peer nomination method was used to assess peer status [45,46]. The participants had a class list in front of them with the first names of their class’s participating students and a number for each first name, which was randomly assigned to the students in advance. In the online questionnaire, participants found only the numbers and clicked on the numbers that corresponded to the desired classmates on their class list. The risk of errors was reduced by simply clicking on numbers [93], and the effects of name order [46,94] were reduced by randomizing the numbers for each nomination.

Following Coie et al. [40], who noted that likeability and peer rejection were not opposite ends of the same continuum, and to follow Marks et al.’s [47] recommendations for popularity and unpopularity, we measured the four dimensions separately. For this purpose, the adolescents were asked to nominate anonymously those classmates whom they “like the most” and those whom they “like the least” with the following instruction: “Click on the numbers assigned to your classmates on the class list. Do not click on your own number.” For popularity and unpopularity, the adolescents were asked to nominate the classmates on the list whom they thought were popular and unpopular on separate items with the same instruction. The sum of the respective nominations that each adolescent received from their peers was used to derive individual scores. The scores were standardized within each class. Thus, prior to the LPA, no categorical classification into commonly used status groups (e.g., social preference or social impact) was made, as belonging to a category would preclude the formation of peer-status profiles.

#### 2.2.3. Covariates

Depression and anxiety. Using 24 items from the Hopkins Symptom Checklist [95], depression and anxiety were captured as symptoms (Cronbach’s α = 0.96). The items were rated on a four-point Likert scale from 1 = not at all to 4 = extremely. Higher scores indicated a higher severity of anxiety and depression symptoms. Due to the participants’ young age (12–13 years old), the item “loss of sexual interest or pleasure” was excluded from the original scale version with 25 items. The mean score per student was calculated for the LPA.

Dissociation. Dissociation was measured using a short scale from the existing Dissociation Tension Scale (DSS) acute [96], which is used to assess dissociative symptoms as a disturbance or discontinuity of consciousness [97]. One item each on analgesia (changes in sensory processes), somatoform (sensory and motor disturbances), depersonalization (feelings of unreality in relation to self), and derealization (feelings of unreality in relation to the environment) composed the DSS-acute. Participants rated on a four-point Likert scale with items ranging from 1 = not at all to 4 = very strongly (Cronbach’s α = 0.85); items included, “my body feels like it does not belong to me” or “people or things around me do not seem real.” A mean score for each student was calculated for the LPA.

Assigned sex. Assigned sex was obtained from school class lists in which adolescents were categorized as male = 0 or female = 1.

Socio-economic status. Information on the adolescents’ socioeconomic status proves to be difficult because only a few adolescents have knowledge about their parents’ professions or even the income. Therefore, Broer et al. [98] recommend several indicators in the form of a composite score. Following Kassis et al. [11], the present study used adolescents’ sociocultural status as a composite score for students’ socioeconomic background with the dimensions of education- and computer-related possessions, parents’ education level, and number of books in the household (Cronbach’s α = 0.71). A total score was formed from the three scales and divided into the expressions low = 1, medium = 2 and high = 3.

Migration background. The definition of people with a migration background depends on the context and migration policy because different rights and obligations create different contexts. In Switzerland, according to the Federal Statistical Office, the population with a migration background includes: “all foreign nationals, naturalized Swiss citizens, except for those born in Switzerland and whose parents were both born in Switzerland, as well as Swiss citizens at birth whose parents were both born abroad” [99]. Therefore, we conceptualized migration background as follow: If the adolescents or their parents did not have Swiss nationality or if adolescents were not born in Switzerland, they had a migration background (=1). If the above characteristics did not apply, they did not have a migration background (=0).

### 2.3. Analysis Plan

To answer the first Research Question 1 (RQ1) and test Hypothesis 1 (H1), LPA was used to identify unobserved heterogeneous profiles with four continuous indicators (likeability, rejection, popularity, and unpopularity) in two groups consisting of adolescents with and without parental physical abuse experiences. *t*-tests were conducted in both groups to analyze the differences among the four indicators. LPA identifies groups or types of people who exhibit different profiles of personal and/or environmental characteristics [100]. Compared to variable-centered analyses, LPA allows for a closer look at profiles and their predictors as well as a distinction between groups that are revealed [101]. Distinct from latent class analysis, LPA includes continuous indicators to identify different groups in empirical data [102]. To determine the number of profiles, an iterative process was chosen in which one to six profile solutions were tested to determine the optimal number of profiles.

A series of LPAs were conducted for the two groups—abuse (experiences of parental physical abuse) and no abuse (no experience of parental physical abuse) to assess the accurate number of profiles for both groups. The appropriate model was chosen based on the following criteria: Bayesian Information Criteria (BIC), Akaike Information Criteria (AIC), the Sample-Adjusted BIC (SABIC), the Bootstrap Likelihood Ratio Test (BLRT), the (adjusted) Lo–Mendell–Rubin Test (LMR and aLMR) posterior classification probabilities, and entropy value. The model better fits the smaller values of AIC, BIC, and SABIC [102,103]. Based on the power of the selection criteria and the different sample sizes for adolescents with parental physical abuse experiences (*n* = 394) and youth without parental physical abuse experiences (*n* = 1565), the focus was put on LMR, aLMR, and BIC [104], although all selection criteria were considered. LMR, aLMR, BLRT, and BIC are considered stable criteria for numbers of profiles regardless of sample size, whereas the entropy value and AIC do not seem to be as reliable for decisions of profile numbers [104]. The LMR and BLRT tests’ significant *p*-values indicate that the fit of a model with k-classes improves significantly compared to the previous model with k-1 classes [103]. Classification diagnostics further support the class enumeration process, where the classification probabilities for the most likely latent class membership represent the probability that an individual is part of a specific latent class. Maysn [105] considers values greater than or equal to 0.70 as desirable.

All analyses were conducted in Mplus version 8.4 [106] with maximum likelihood estimation and robust standard errors due to non-normal distributions. Missing data were estimated using the Full Information Maximum Likelihood (FIML) method. Random starts were increased to 1000 and final optimizations to 100 to avoid local solutions [101]. All models were estimated using the default setting of Mplus and no cases were excluded due to the exploratory character of the underlying research questions [100].

In a second step, to determine whether the LPA profiles and parameters (mean values comparison) significantly differed from each other, a series of pairwise Wald tests were conducted for the two groups (abused vs. non-abused adolescents).

To answer the second Research Question 2 (RQ 2) and test Hypothesis 2 (H2), we tested measurement invariance (MI). The separate LPAs for the two groups were compared to evaluate whether the latent profiles’ number and nature were the same across the two groups. Non-invariance would mean that the profiles in the abuse and no abuse groups were characterized unequally; therefore, not directly comparable and interpretable [107], which results in further analysis that must be performed separately across groups [108].

To answer the third Research Question 3 (RQ3) and test Hypothesis 3 (H3), a three-step approach for auxiliary variables with the Mplus R3STEP [109] auxiliary command was conducted to predict the profile membership. We examined whether depression and anxiety symptoms, dissociation, assigned sex, socioeconomic status, and migration background were related to a higher probability of adolescents belonging to one specific profile rather than another. This method was corrected for a classification error [109].

## 3. Results

### 3.1. Descriptive Statistics

*T*-tests were conducted (see Table 1) to analyze the four indicators’ differences in both groups. We found a small significant effect only for the indicator of rejection; otherwise, no effects on the measures were detected. Despite the homogeneous mean values in three out of four indicators in both groups, we expected that the profiles of the person-centered LPAs would differ in terms of indicators. The prevalence of physical abuse was 20.1%.

### 3.2. Research Question 1: Latent Profiles of Peer Status

Before employing the LPA, bivariate correlations between the peer status variables were checked (see Table 2). To examine the number of peer-status profiles and their characterizations, the optimal number of profiles was selected to determine whether the same number of profiles could be found in each group. We defined two separate LPA models for this purpose. The model fit indices for each latent profile model were analyzed separately for the groups *abuse* and *no abuse* (see Table 3).

For the *abuse* group, the AIC, BIC, and SABIC values increased from the one-profile solution to the six-profile solution, indicating the fit was reproduced better with each subsequent profile model. The *abuse* group showed a significant LMR, aLMR, and BLTR test from the two-profile solution to the three-profile solution, but not from the three-profile solution to the four-profile solution. The entropy value decreased considerably from the three-profile solution (0.89) to the four-profile solution (0.80), which supported the rejection of the four-profile solution. Furthermore, one class proportion in the four-profile solution was only 4% (*n* = 15) and could therefore reduce the profile’s accuracy [100]. Classification probabilities for the most likely latent class membership are satisfactory with values above 0.7. These considerations argued for a three-profile solution as the most parsimonious solution for the *abuse* group. Figure 1 displays a plot with the three-profile model for the subsample with parental physical abuse experiences.

The first profile in the three-profile solution shows a group of adolescents whose peers liked them, but these adolescents otherwise received low scores. Therefore, this proportionally biggest profile was named *liked* (*n* = 318, 80.7%). The second profile was named *liked-popular* (*n* = 45, 11.4%) because it displayed a group of adolescents who were liked in their class and their peers considered popular. The third profile, *rejected-unpopular* (*n* = 31, 7.8%) comprises adolescents whose classmates rejected them and were nominated as unpopular.

In the *no abuse* group, the *p*-value of LMR, aLMR, and BLTR tests showed that a four-profile solution was more optimal compared to a five-profile solution LMR and aLMR no longer provided a significant solution. The class proportion of 1% (*n* = 23) was not sufficient in the five-profile solution and was therefore rejected. Here, values above 0.7 also proved to be satisfactory for classification probabilities for the most likely latent class membership. Based on these considerations, we decided that the four-profile solution indicated the best fit and was the most parsimonious model for the *no abuse* sample (Table 2). Three profiles were named the same in both samples because they had very similar characteristics in relation to the indicators.

Figure 2 shows a plot with the four-profile model for the subsample without parental physical abuse experiences.

The first profile was named *liked* (*n* = 1071, 68.4%), the second profile was termed *liked-popular* (*n* = 108, 6.9%), and the third profile displayed *rejected-unpopular* adolescents (*n* = 72, 4.6%) because the indicators showed similar levels of mean values as in the *abuse* group. The fourth profile was named *average* (*n* = 314, 20%) because these adolescents had average levels on the indicators liked, rejected, and unpopular and had similar levels on the indicator popular as adolescents in the liked profile.

### 3.3. Research Question 2: Comparison of LPA Profiles

To investigate the differences in the underlying profiles of peer status for adolescents with and without parental physical abuse experiences, we considered measurement invariance. In the current study, measurement invariance was not given and did not need to be tested further because the number of profiles differed between the two groups (three-profile solution for the *abuse* group and four-profile solution for the *no abuse* group). A lack of measurement invariance means that the two groups must be considered independently, and further analyses and interpretation must be performed separately [108].

To determine whether the profiles in the separate models generally differed from each other, we conducted a Wald test. This revealed an overall significance of the *abuse* model χ^2^ (8) = 267.14, *p* < 0.001 and the *no abuse* model χ^2^ (12) = 1315.33, *p* < 0.001. Thus, the profiles in each model differed from each other. Table 4 presents all pairwise comparisons.

#### 3.3.1. Pairwise Comparison in the No Abuse Model

The mean values of the indicator likeability differed in all three status profiles. The rejection indicator mean level in the *no abuse* model differed significantly between the *rejected-unpopular* profile and the other three profiles. However, there was no significant difference found in the rejection indicator mean level between the *liked* and the other two profiles, while the mean levels differed between the *liked-popular* and the *average* profiles.

For the popularity indicator’s mean values, only the *liked-popular* profile differed significantly from the other three profiles, while no difference was found in those other three profiles. The results were entirely different for the unpopularity indicator’s mean levels, which differed significantly between all four profiles.

#### 3.3.2. Pairwise Comparison in the Abuse Model

In the *abuse* model, the likeability mean levels of the profiles *liked* and *liked-popular* differed from the mean values of the *rejected-unpopular* profile. However, the likeability indicator’s mean level did not differ significantly between the two profiles *liked* and *liked-popular*. The same picture emerged for the rejection indicator’s average values.

The popularity indicator’s mean levels differed significantly from the *liked-popular* profile to the other profiles but not between the *liked* and *rejected-unpopular* profiles. There was a significant difference between the unpopularity indicator’s mean levels between the *rejected-unpopular* profile and the other two profiles but not between the *liked* and the *liked-popular* profiles.

### 3.4. Research Question 3: Predictors of Latent Profile Membership

To investigate the extent to which different internalizing symptoms predicted peer-status profiles, a multinomial logistic regression was performed using the automatic three-step procedure of Mplus (R3STEP). This allowed including the predictors in both groups separately (see Table 5). This also allowed assessing depression, anxiety, and dissociation as internalizing symptoms as well as gender, socioeconomic status, and migration background as sociodemographic covariates predicting latent profile membership.

#### 3.4.1. Internalizing Symptoms Variables

In the *abuse* group, the chances decreased of adolescents being in the *liked* rather than in the *rejected-unpopular* profile with increasing dissociation symptoms. With increasing anxiety, the chances decreased of adolescents in the *no abuse* group being in the *liked* or the *liked-popular* profile rather than being in the *rejected-unpopular* profile. In addition, with increasing anxiety, the chances increased of adolescents being in the *liked* or *rejected-unpopular* profile rather than the *average* profile. No significant differences were found in the *abuse* group regarding depression and anxiety. In the *no abuse* group, no significant differences were found for depression and dissociation.

#### 3.4.2. Sociodemographic Variables

In the *abuse* group, adolescents with higher socioeconomic status in comparison to adolescents with lower socioeconomic status had a higher probability of being in the *liked* profile than in the *rejected-unpopular* profile. Adolescents with a migration background in comparison to native youth had a higher probability of being in the *liked-popular* profile than the *rejected-unpopular* profile. No other significant comparisons were found in the *abuse* group. In the *no abuse* group, females were more likely than males were to be in the *liked* profile and *average* profile than in the *rejected-unpopular* profile. On the other hand, compared to females, males were more likely to be in the *liked-popular* profile than in the *liked* profile. In the *no abuse* group, no significant profile differences were found relating to migration background and socioeconomic status.

## 4. Discussion

With about a 20% prevalence, the present study confirms the alarming international finding that one in five adolescents in Switzerland experience parental physical abuse [3,5]. The present study aimed to find out whether distinct forms of peer status emerged in adolescents with and without parental physical abuse experience. Using the resilience framework as well as the interactional model, the following research questions were stated: How many peer-status profiles can be found for adolescents with and without parental physical abuse experiences, and how are they characterized? Are there differences in the underlying profiles of peer status between adolescents with and without parental physical abuse experiences? How do different forms of internalizing symptoms (depression, anxiety, and dissociation) predict the memberships of these underlying peer-status profiles?

As a first result, two profiles were found for the two groups of adolescents (with and without abuse experiences). The second hypothesis, which expected that the peer-status profiles of adolescents with and without abuse experiences would differ, was confirmed. Peers indeed perceived differently the four dimensions of perceived peer status.

In the group of adolescents with parental physical abuse experiences, we uncovered three peer-status profiles: *liked*, *liked-popular*, and *rejected-unpopular*. Thus, there were differences in peer-status profiles depending on physical abuse experiences. We uncovered the additional profile *average* in the *no abuse* group. Van den Berg et al. [37] also found four similar clusters for grade 8 youth, namely *liked*, *popular*, *unpopular-disliked*, and *average*. Therefore, a very similar picture emerged in our analysis, except that we found a *liked-popular* group instead of a *popular* group. For grade 7, van den Berg et al. [37] found three clusters, namely *popular-liked*, *unpopular-disliked*, and *average*. A possible explanation for the diverging results could be that the adolescents were still in grade 7, while the *popular* group might appear in grade 8. Furthermore, it may also be because these adolescents had just entered secondary school at the time of data collection and the peer group needs time to form dynamics and establish peer status. Our first hypothesis, which expected at least three profiles to be *rejected-unpopular*, *liked-popular*, and *average*, was thus confirmed only for the group of adolescents without abuse experience. However, it was not confirmed for the group of adolescents with abuse experiences because they did not display an *average* profile.

In particular, peer rejection played an important role for peer status and abuse experiences, both by showing significant differences between the two abuse groups, as the t-test indicated, and by accounting for the profiles that were found within the two groups. Older studies have indicated that adolescents tend to be less simultaneously popular and well liked, which an increased potential for aggression among popular adolescents has explained [41,110]. This was confirmed in our study because for adolescents with and without abuse experiences, popular and liked formed the smallest profile. However, interestingly, this profile was larger among the adolescents with abuse experiences. Thus, the question arises whether *liked-popular* adolescents with abuse experiences represent a substantively different group than *liked-popular* adolescents without abuse experiences. For future research, it would be interesting to explore how the profiles of the two groups differ regarding content.

With respect to adolescents’ parental physical abuse experiences, the *rejected-unpopular* profile is particularly important to consider in future research in relation to peer victimization and peer aggression because abused children appear to show increased aggression toward peers [111]. From a psychological perspective and according to resilience theory, peer rejection might be considered a risk factor for adolescents’ adaptive development [112]. Therefore, it can be assumed that the adolescents in the *rejected-unpopular* profiles did not undergo resilient development regarding peer relationships. One possible explanation could be, as Martin-Babarro et al. [113] hypothesized, that a lack of a supportive environment in families experiencing abuse might compromise building resilience. To date, research on peer relationships has focused on sociological and educational perspectives, although a resilience theory perspective could potentially provide meaningful information on protective factors for youth who struggle with peer rejection [114].

From a social learning perspective, peer rejection is an elicited environmental response to the child’s behavior [115]. Based on this, it would be possible that youths in the *abuse* group were more likely to be conspicuous via aggressive behavior, which increases the chances of peer rejection [53]. The fact that youth who have experienced abuse are more likely to experience peer rejection is reflected in the fact that the *rejected-unpopular* profile was twice as large, relatively speaking, as the *rejected-unpopular* profile without abuse. Based on our findings, peer status cannot be considered generally applicable within a school class, but this status might depend on various factors. Therefore, in addition to physical abuse experiences, it would be interesting to consider other risk and protective factors for the construction of latent peer-status profiles.

However, our profiles differed from van den Berg et al.’s [37] profiles in that we did not find an *average* profile in the *abuse* group, but instead identified a *liked* profile. The *liked* profile contained the largest proportion (80.7%) of adolescents in the *abuse* group and consisted of youths with above-average like levels from their peers, but very few nominations for the other three indicators, and thus, were neither popular nor unpopular. Analogous to van den Berg et al. [37], no status group was found that consisted of popular and rejected adolescents. Older studies that had a significant relationship between popularity and rejection found *popular-rejected* groups, e.g., [41]. One possible explanation could lie in the current study’s and that of van den Berg et al.’s [37] person-centered approaches, which seem to differentiate more than variable-centered methods do. Moreover, with increasingly complex survey and evaluation procedures in the sociometric field, identified status groups may change.

As a further finding, the present study derived unique associations between internalizing symptoms and peer status in adolescents with and without parental physical abuse experiences. In the *abuse* group, dissociation as an internalizing symptom significantly increased the likelihood of belonging to the *rejected-unpopular* profile compared to the *popular* profile. This confirms hypothesis 3 because we expected that the development of dissociative problems would often be a consequence of abuse, especially after physical abuse [18]. Abused children are also more likely to exhibit attention deficits and insufficiencies in emotion regulation, which manifest in emotional lability, negativity, and contextually inappropriate expressions of emotions, in turn leading to problems in interpersonal relationships [111,116]. Rejection from peers can in turn lead to increased dissociation because painful peer rejection, although not considered a major trauma, is nonetheless associated with dissociation in children [81]. Therefore, it is not surprising that adolescents with parental physical abuse experiences displaying dissociations are more likely to experience peer rejection and to be seen as unpopular. Considering recent research shows a link between high levels of dissociation and the frequency and severity of self-harming behavior in adolescents [117], prevention policies should focus on youth in the *rejected-unpopular* profile with higher levels of dissociation.

Unexpectedly, depressive symptoms did not predict profile membership in the *abuse* group, although we expected depression to predict membership in the *rejected-unpopular* profile [73]. An explanation might be that depression is not directly related to peer rejection [118]. Another possible explanation could be that depression appears to be more prevalent in other forms of exposure to abuse, such as emotional abuse, and thus could show effects related to peer status group membership. For example, Humphreys et al. [119] and Gardner et al. [66] found in their meta-analysis that there was a higher correlation between depression and emotional abuse than there was with physical abuse.

In the *no abuse* group, anxiety as an internalizing symptom played a significant role as a predictor for profile membership in comparison to depression or dissociation, which did not predict profile membership. Adolescents who displayed higher anxiety levels were more likely to be in the *rejected-unpopular* or *liked* group than in the other profiles. Although the literature has associated peer preference with a lower risk of developing internalizing behaviors [35], our person-centered analysis using the four status dimensions shows that this is only partially confirmed. In our case, anxious adolescents without parental physical abuse experiences were more likely to be in either the *rejected-unpopular* or the *liked* profiles. This can possibly be explained by the fact that likeability and rejection are summed up in the peer preference construct, which the loss of nuances of the individual dimensions accompanies. Thus, the results might contribute to the assumption that peer acceptance in particular should be operationalized with two separate dimensions.

On the one hand, this supports findings from previous studies that revealed that rejected adolescents showed anxiety more often than liked, average, or controversial adolescents [72]. On the other hand, adolescents with elevated anxiety levels were also more likely to be in the liked profile, which is in line with the Baartmans et al.’s [75] findings. In that study, anxious children experienced peer rejection less than did children with lower social anxiety levels. Among adolescents who did not experience parental physical abuse, increased anxiety levels were particularly associated with psychological control and harsh parental control [120]. Future research should include information on parenting practices and styles to determine what underlying mechanisms link increased anxiety levels and peer status of adolescents who do and do not experience abuse. It seems like anxiety has more of an effect on popularity than acceptance does, although more in-depth analysis on this would be needed in the future to make accurate statements.

Regarding the sociodemographic predictors, consistent with previous studies, we found no link between gender and peer status in early adolescence in the *abuse* group [23,82]. By contrast, in the group of adolescents who did not experience parental physical abuse, we identified significant gender differences. Female gender was predictive for the membership in the *liked* profile and *average* profile compared to the *rejected-unpopular* profile, whereas male gender predicted membership in the *liked-popular* profile compared to the *liked* profile. These results differ from van den Berg et al.’s [37] findings, which showed that male participants were more likely and overrepresented in the *rejected-unpopular* group in grades 7 and 8. Our results argue for the “backlash effect” [121], which states that there exist higher requirements for niceness that apply to women than to men. According to van den Berg et al. [44], this could result in likeability and popularity correlating less strongly in girls because of gender stereotypes. Gender norms for likeability (associated with niceness) and popularity (associated with dominance and influence) may explain that male adolescents in the present study showed higher odds of being in the *liked-popular* group, and female adolescents had a higher chance of being in the *liked* or *average* group.

Further, adolescents with physical abuse experience and with a migration background had a higher probability of being in the *liked-popular* profile than in the *rejected-unpopular* profile. These results support Kovacev and Shute’s [84] and Stevens et al.’s [85] previous findings, which showed that immigrant youth received high peer acceptance scores as well as high popularity scores, especially if they had positive attitudes toward the heritage and the host cultures. Moreover, similar to Bukowski et al. [86], high socioeconomic status significantly predicted profile membership in the *liked* profile compared to the *rejected-unpopular* profile. This finding is partly in line with Alivernini et al. [83], who found that low socioeconomic status predicted lower peer acceptance scores. However, these results must be interpreted with caution, considering that in the present study, socioeconomic status was operationalized as sociocultural capital without information about the parents’ income. Interestingly, migration background and socioeconomic status did not predict profile membership in the *no abuse* group.

### 4.1. Limitations

The present study generated some important findings and had several important strengths, such as a large sample including adolescents recruited from the general population rather than just a clinical sample. Nevertheless, we need to address a few limitations. First, cross-sectional data were used to examine the profiles presented here, and it was not possible to assess the relative timing of maltreatment and the emergence of the internalizing symptoms. Therefore, to test the profiles’ stability as well as to draw causal conclusions, longitudinal data with three waves are also needed to determine how internalizing symptoms actually associate with profile membership and how much of the internalizing symptomatology causes profile membership. Second, abuse often co-occurs with other adverse childhood experiences [122], such as other forms of parental abuse, which were not systematically considered in the present study and whose effects we were unable to separate from parental physical abuse. Therefore, the results need to be interpreted cautiously and cannot be generalized for different forms of parental abuse. Third, compared to many studies, valid peer nominations have been used to obtain sociometric data on peer acceptance and peer popularity [123,124]. This method has significant advantages over self-reports [125], but does not rule out the possibility that considering the combination of sociometric data, self-reports, and teacher data could increase the reliability of peer status and lead to more accurate peer-status profiles. Further, the terminology of popularity is understood differently depending on the cultural contexts [46]. This must be considered when interpreting the results regarding popularity. Fourth, this study’s sample was based exclusively on data from Swiss adolescents. In Switzerland, after entering secondary school, adolescents usually spend their school years in the same classes with the same peer groups for at least 3 years. Thus, the peer group is not mixed with other school classes or grades, which may provide only limited insight into the role of peer status in other ethnic, cultural, and educational contexts. Finally, the dichotomization of physical abuse as a grouping variable in the LPA did not fully do justice to the severity of the physical abuse experience because no nuances within the abuse group could be considered.

### 4.2. Future Research Directions

Positive peer relationships are protective factors regarding parental physical abuse experiences [126]. From the resilience framework perspective, the high percentage of future resilience research should focus on the factors that promote peer acceptance and popularity in classrooms. Peer acceptance and popularity in turn could be considered as protective factors for individuals’ self-concepts [127]. Because there is limited person-centered research on these protective factors and peer popularity seems to have differing effects [57], this topic should be expanded in future research. We recommend that researchers replicate our findings in cross-cultural studies. In addition, to gain a more differentiated insight into the youth groups in the peer-status profiles, it would be beneficial for future researchers to closer examine the sociodemographic variables. As Kassis et al. [128] showed, an intracategorical and intersectional approach to gender identity and sexual attraction offers a picture that is much more differentiated of the psychological state of early adolescents than the binary categorization of female and male is. Especially regarding likeability and popularity, a more diverse picture would be interesting, as most research is based on a binary distinction.

## 5. Conclusions

The present study provided valuable insights into the role of experienced parental physical abuse on adolescents’ positions within the peer group membership. Peer status should be involved in school and classroom interventions and should be considered as a protective and a risk factor in relation to experiences of parental abuse and violence resilience. This could include trauma-informed training for teachers, because youths who have experienced maltreatment are 2.7 times more likely to be diagnosed with a mental illness compared to their non-abused peers [129]. The peer group and peer status, in particular peer rejection, as part of the system in which adolescents are embedded can play a crucial role for adolescents who bear such a burden of traumatic experience and should be further considered in future resilience research. Dissociation as a severe trauma response plays an important role in relation to the position within the peer group. Thus, especially with regard to adolescents who experience physical abuse, a focus should be placed on dissociative symptoms and not only on depression and anxiety as internalized symptoms, which is mainly the case in research. Therefore, students with dissociative symptoms and a low peer status should be closely monitored as an especially vulnerable group of individuals.

## Figures and Tables

**Figure 1 children-09-00599-f001:**
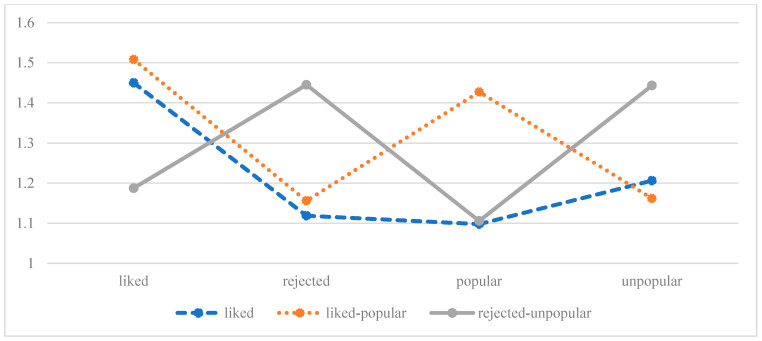
Three profile solution, abuse group.

**Figure 2 children-09-00599-f002:**
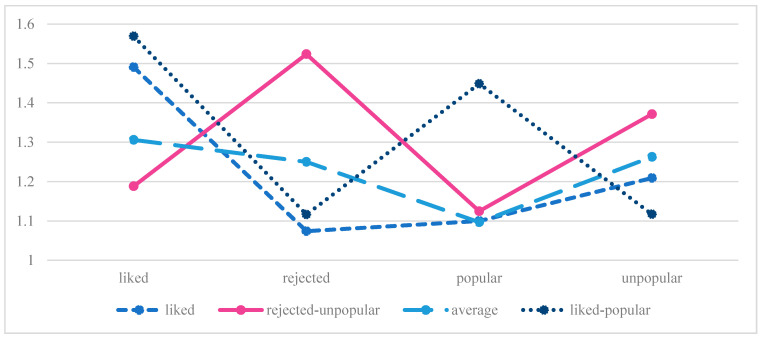
Four profile solution, no abuse group.

**Table 1 children-09-00599-t001:** Descriptive statistics, sample mean levels (and standard deviations) of all observed variables (abuse *n* = 394, no abuse *n* = 1565) and effect sizes (Hedges’ g).

Variable	Mean (SD)		t	g
	Abuse	No Abuse		
Likeability	1.43 (0.17)	1.44 (0.17)	0.971	-
Rejection	1.15 (0.15)	1.14 (0.14)	−1.98 *	0.07
Popularity	1.14 (0.14)	1.13 (0.13)	−1.30	-
Unpopularity	1.22 (0.15)	1.22 (0.14)	−0.09	-
Depression	2.05 (0.64)	1.81 (0.63)		
Anxiety	2.00 (0.78)	1.66 (0.65)		
Dissociation	1.61 (0.73)	1.31 (0.54)		

* *p* < 0.05.

**Table 2 children-09-00599-t002:** Bivariate correlations peer status, spearman.

	Likeability	Rejection	Popularity	Unpopularity
Likeability	1	−0.567 **	0.212 **	−0.186 **
Rejection		1	0.028	0.278 **
Popularity			1	−0.194 **
Unpopularity				1

** *p* < 0.01.

**Table 3 children-09-00599-t003:** Model fit indices for latent profile analysis of adolescents with and without parental physical abuse experience, 1–6 profiles.

	Nr. of Profiles	AIC	BIC	ABIC	Entropy	LMR LR Test	ALMR LR Test	Smallest Class %	BLRT	Classification Probabilities
abuse	1	−1472.45	−1440.64	−1466.02						
	2	−1662.20	−1610.50	−1651.75	0.89	0.14	0.14	11%	<0.001	0.99; 0.83
	3	−1760.54	−1688.97	−1746.08	0.89	<0.01	<0.01	8%	<0.001	0.98; 0.87; 0.81
	4	−1825.98	−1734.52	−1807.50	0.80	0.17	0.17	4%	<0.001	0.83; 0.91; 0.90; 0.91
	5	−1870.55	−1759.21	−1848.06	0.84	0.17	0.17	3%	<0.001	0.85; 0.84; 0.93; 0.86; 0.96
	6	−1909.78	−1778.56	−1883.27	0.84	<0.05	<0.05	3%	<0.001	0.88; 0.98; 0.89; 0.95; 0.88; 0.03
no abuse	1	−6561.20	−6518.35	−6543.76						
	2	−7395.26	−7325.64	−7366.93	0.89	<0.001	<0.001	14%	<0.001	0.98; 0.87
	3	−7815.38	−7718.98	−7776.16	0.89	0.01	0.01	8%	<0.001	0.97; 0.87; 0.85
	4	−8025.13	−7901.95	−7975.02	0.84	<0.05	<0.05	7%	<0.001	0.95; 0.89; 0.79; 0.84
	5	−8224.50	−8074.54	−8163.49	0.86	0.11	0.11	1%	<0.001	0.94; 0.82; 0.91; 0.90; 0.83
	6	−8334.10	−8157.36	−8262.20	0.86	0.32	0.32	1%	<0.001	0.82; 0.94; 0.91; 0.94; 0.80; 0.78

AIC = Akaike information criterion; BIC = Bayesian information criterion; ABIC = sample-size adjusted BIC; LMR LR = Vuong–Lo–Mendell–Rubin Likelihood Ratio Test; ALMR LR = Lo–Mendell–Rubin Adjusted LRT Test; BLRT = bootstrap likelihood ratio test; CP = Classification Probabilities for the Most Likely Latent Class Membership.

**Table 4 children-09-00599-t004:** Wald Test, means and standard errors of the profiles.

Variable	Sample	1 *Liked* M (*SE*)	2 *Liked-Popular* M (*SE*)	3 *Rejected-Unpopular* M (*SE*)	4 *Average* M (*SE*)
Likeability	abuse	1.450 (0.013) ^3^	1.508 (0.029) ^3^	1.187 (0.040) ^1,2^	-
	no abuse	1.491 (0.008) ^2,3,4^	1.569 (0.024) ^1,3,4^	1.188 (0.019) ^1,2,4^	1.306 (0.010) ^1,2,3^
Rejection	abuse	1.119 (0.010) ^3^	1.156 (0.024) ^3^	1.445 (0.076) ^1,2^	-
	no abuse	1.074 (0.004) ^3^	1.116 (0.022) ^3,4^	1.524 (0.022) ^1,2,4^	1.250 (0.016) ^2,3^
Popularity	abuse	1.098 (0.008) ^2^	1.427 (0.040) ^1,3^	1.106 (0.023) ^2^	-
	no abuse	1.100 (0.005) ^2^	1.449 (0.032) ^1,3,4^	1.125 (0.021) ^2^	1.097 (0.008) ^2^
Unpopularity	abuse	1.206 (0.009) ^3^	1.162 (0.025) ^3^	1.443 (0.068) ^1,2^	-
	no abuse	1.209 (0.004) ^2,3,4^	1.117 (0.011) ^1,3,4^	1.371 (0.030) ^1,2,4^	1.263 (0.011) ^1,2,3^

Abuse = parental physical abuse; no abuse = no parental physical abuse; ^1,2,3,4^ indicate significant Wald Test to the respective profile.

**Table 5 children-09-00599-t005:** Multinomial logistic regression of socio-demographic covariates, depression, anxiety, and dissociation to the identified latent profile membership: parameter estimates of both models.

Reference Class		Rejected-Unpopular vs. Liked		Rejected-Unpopular vs. Liked-Popular		Liked vs. Liked-Popular		Average vs. Liked		Average vs. Liked-Popular		Average vs. Rejected-Unpopular	
	Predictor	Estimate (*SE*)	*OR*	Estimate (*SE*)	*OR*	Estimate (*SE*)	*OR*	Estimate (*SE*)	*OR*	Estimate (*SE*)	*OR*	Estimate (*SE*)	*OR*
abuse	Male	0.016 (0.633)	1.016	0.559 (0.778)	1.750	0.544 (0.494)	1.723	-		-		-	
	Migration Background	1.160 (0.661)	3.189	**2.214 *** (1.009)	9.152	1.054 (0.774)	2.870	-		-		-	
	High Socio-economic Status	**0.706 *** (0.334)	2.025	0.530 (0.486)	1.699	−0.176 (0.383)	0.839	-		-		-	
	Depression	−0.781 (0.507)	0.458	−1.452 (0.824)	0.234	−0.670 (0.713)	0.512	-		-		-	
	Anxiety	1.807 (0.965)	1.807	2.181 (1.157)	8.853	0.373 (0.614)	1.453	-		-		-	
	Dissociation	**−1.002 *** (0.448)	0.367	−0.928 (0.628)	0.396	0.075 (0.443)	1.078	-		-		-	
no abuse	Male	**−0.777 *** (0.321)	0.460	0.225 (0.414)	1.252	**1.001 ***** (0.284)	2.722	**−0.483 **** (0.187)	0.617	0.518 (0.313)	1.679	0.294 (0.366)	1.342
	Migration Background	−0.210 (0.284)	0.811	0.278 (0.371)	1.320	0.487 (0.266)	1.628	−0.133 (0.183)	0.876	0.355 (0.288)	1.426	0.077 (0.324)	1.080
	High Socio-economic Status	−0.216 (0.228)	0.805	−0.489 (0.293)	0.613	−0.272 (0.201)	0.762	−0.117 (0.135)	0.889	−0.390 (0.218)	0.677	0.099 (0.261)	1.104
	Depression	0.080 (0.422)	1.084	−0.138 (0.572)	0.871	−0.218 (0.409)	0.804	−0.141 (0.238)	0.868	−0.359 (0.436)	0.698	−0.222 (0.479)	0.801
	Anxiety	**−0.754 *** (0.368)	0.470	**−1.215 *** (0.529)	0.297	−0.461 (0.411)	0.631	**0.664 ***** (0.245)	1.943	0.203 (0.439)	1.225	**1.418 ***** (0.428)	4.131
	Dissociation	0.122 (0.364)	1.130	0.654 (0.495)	1.924	0.532 (0.360)	1.702	−0.206 (0.214)	0.814	0.326 (0.376)	1.385	−0.328 (0.412)	0.720

Estimate = *β* from R3STEP analysis; * *p* < 0.05; ** *p* < 0.01; *** *p* < 0.001.

## Data Availability

The authors will make available the raw data supporting this article’s conclusions without undue reservation, upon the project’s completion in 2023.

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
