# Peer review of "Peer Status as a Potential Risk or Protective Factor: A Latent Profile Analysis on Peer Status and Its Association with Internalizing Symptoms in Adolescents with and without Parental Physical Abuse Experience"

_children, 2022, doi:10.3390/children9050599_

Round 1

Reviewer 1 Report

The text concerns an important problem of the relationship between the fact of being a victim of physical violence used by a parent / parents against a school-age child (adolescents were examined) and its status in the school class and internalization symptoms - especially from the perspective of the psychology of human development. The internalization symptoms included anxiety, depression and dissociation. The study also took into account the gender and socio-economic status of the family, including membership in the migrant environment (which?)

I did not find any clear information whether the study was made of young people from an ethnic minority representing an ethnic group living in Switzerland for some time, or about current migrants? I wonder if the result indicating a positive attitude of students towards colleagues from ethnic minority environments can be explained, for example, by the need for social approval, or is it the effect of effective education integrating young people, openness to "others"? It would also be useful to explain some results to use other indicators of socio-economic status - the authors are aware of this. Perhaps they can complete the data (?)

The study is interesting. Preceded by an introduction, which refers to the results of previous research - conducted by other authors. Researchers emphasize the usefulness of the concept of resilience. As a result of the analyzes, they identified profiles illustrating the degree of acceptance / sympathy/likeability and popularity as well as the lack of popularity – unpopularity, and rejection by classmates. In explaining the results, they refer, inter alia, to gender stereotypes.

They created a model of relationships between: the (independent) bivalent explanatory variable: being a victim of physical violence used by parents vs the lack of disclosed violence used by parents and the dependent variable which is the social status in the group of students from the class (the so-called profile). The internalizing symptoms (dissociation, depression and anxiety) - which were treated as consequences of parental abuse - were given the importance of predictors of the student's social status. This group (of predictors) was extended to include such variables as gender, socio-economic status and migration origin.

It is a logical sequence of associations. In explaining the results, the researchers referred to clear cause-and-effect relationships. I take this article as information about the initial stage of research on this problem. It would be worthwhile - in subsequent studies - to pay attention to the scale of violence, its duration (as noted by the authors), the perpetrator (mother, father, both and consider the importance of the relationship: gender of the parent and child), to explain the mechanisms leading to the symptoms and severity symptoms. In explaining the aggression occurring in some of the respondents (from the abused group), one can refer to modeling, imitation, displaced aggression, efforts to increase self-esteem, defend the self-esteem suppressed by the parent, and other mechanisms. Due to the school environment, it is justified to take into account the level of educational achievement of the student, which may be important for his position in the classroom.

I believe that researchers still face the challenge of explaining the variation in status (acceptance, rejection and popularity profile) in a group of students who are not physically abused but exhibit symptoms of depression, dissociation and severe anxiety / anxiety.

It would be valuable to fully enrich the sociological and pedagogical perspective that seems to dominate the approach to the problem - with a psychological perspective.

The results concerning the relationship between internalization symptoms and the social position of a student from a properly functioning family are also noteworthy - what is their genesis?

Can the authors supplement the text with information on the reliability indicators of the tools used? alfa Cronbach?

The above does not mean that the text is not valuable. It is very inspiring and the results of the research presented in the article may be the starting point for further explorations enriching the knowledge about the psychological and social consequences of physical abuse against a child by parents.

Reviewer 2 Report

This paper presents well-done statistical analyses and a thorough Discussion.  There are some areas that need to be clarified before the paper can be published:

1)  In lines 40-48, there are examples of the effects of parental abuse.  There should be examples given of the effects of specifically parental physical abuse.

2)  In lines 58-61, it states that peer rejection increases attribution of hostile intentions and decreases development of competent solutions.  This is not typically the case.  The child may be abused and develop a view where they perceive interactions as aggressive and attacking or learn poor solutions from poor parental behavior and take those to his/her peer interactions (see Kenneth Dodge's work).

3)  In the section "Peer acceptance and popularity as two distinct aspects of youths' peer status," the authors should state how they generate their categories to categorize peers that they propose in their hypotheses.

4)  The two paragraphs from lines 158 to 173 repeat the content of the paragraphs in lines 64 to 77.  These paragraphs should be merged.

5)  In lines 181 to 183, the authors state (and need a citations to support) that popularity is positively associated with aggressive behavior.  However, other research shows that popularity is associated with prosocial behavior.  The authors should cite that research as well.

6)  In lines 214-215, the authors state that the research they have just presented shows that interpersonal problems can result in depression.  However, none of the research they have presented thus far in the paragraph does support that claim.  This statement should be removed.

7)  In lines 261-262, the authors state that popular girls tend to be less liked.  They reiterate this rationale in the Discussion.  This statement goes against the very definition of popularity as it was originally given by the researchers who developed this typology.  The authors should clarify their thinking here in light of original research.

8)  The statement for hypothesis 3 does not match research question 3.  Research question 3 is about internalizing symptoms as predictors of profiles, while the hypothesis addresses whether physical abuse predicts peer status.

9)  In Section 2.1, it states that 52.6% of the sample were Swiss citizens, meaning that almost half of the sample was from out-of-the country.  Where was the rest of the sample from as far as being immigrants?

10) In Section 2.2, the authors state that their parenting measure assessed corporal punishment as a form of physical abuse.  It is debated in the literature as to whether spanking constitutes physical abuse or not and spanking is not widely accepted as physical abuse internationally.  Thus, the validity of the parenting measure is in question.

11) For the peer status measure, it states that children were asked which students were "popular" and "unpopular."  Were or how were these terms defined?  Students could have interpreted these terms differently, leading to invalid data.

12) The no abuse sample was four times the size of the abuse sample, yielding much more power for the no abuse sample.

13)  In the Limitations section, the generalizability limitations in terms of age range and location (i.e., this study does not generalize to other countries or ethnicities such as Hispanic or African American) should be mentioned.  It should also be mentioned that it may not generalize to other abuse types such as emotional or sexual abuse.

Reviewer 3 Report

interesting approach to look at peer relationships-- the differences were not surprising but the inclusion of the anxious/not abused cohort confusing. Were there other adverse childhood traumas in their lives beside physical abuse
